# Discovery of RNA-binding proteins and characterization of their dynamic responses by enhanced RNA interactome capture

Joel I. Perez-Perri [1], Birgit Rogell[1], Thomas Schwarzl [1], Frank Stein [1], Yang Zhou[1,2,3], Mandy Rettel [1], Annika Brosig [1,2,3,4] & Matthias W. Hentze [1,2]

Following the realization that eukaryotic RNA-binding proteomes are substantially larger than anticipated, we must now understand their detailed composition and dynamics. Methods such as RNA interactome capture (RIC) have begun to address this need. However, limitations of RIC have been reported. Here we describe enhanced RNA interactome capture (eRIC), a method based on the use of an LNA-modified capture probe, which yields numerous advantages including greater specificity and increased signal-to-noise ratios compared to existing methods. In Jurkat cells, eRIC reduces the rRNA and DNA contamination by >10-fold compared to RIC and increases the detection of RNA-binding proteins. Due to its low background, eRIC also empowers comparative analyses of changes of RNA-bound proteomes missed by RIC. For example, in cells treated with dimethyloxalylglycine, which inhibits RNA demethylases, eRIC identifies m6A-responsive RNA-binding proteins that escape RIC. eRIC will facilitate the unbiased characterization of RBP dynamics in response to biological and pharmacological cues.

---

[1] European Molecular Biology Laboratory (EMBL), Meyerhofstrasse 1, 69117 Heidelberg, Germany. [2] Molecular Medicine Partnership Unit (MMPU), Im Neuenheimer Feld 350, 69120 Heidelberg, Germany. [3] Department of Pediatric Oncology, Hematology, and Immunology, Heidelberg University, Im Neuenheimer Feld 430, 69120 Heidelberg, Germany. [4] Faculty of Biosciences, Heidelberg University, 69120 Heidelberg, Germany. Annika Brosig: Candidate for joint PhD degree from EMBL and Faculty of Biosciences, Heidelberg University. Correspondence and requests for materials should be addressed to M.W.H. (email: hentze@embl.de)

RNA-binding proteins (RBPs) play essential roles in gene expression and other cellular functions. Thus their identification and the understanding of their mechanisms of action and regulation is key to unraveling physiology and disease. Recent approaches for the proteome-wide identification of RBPs, especially RNA interactome capture (RIC)[1,2] and modifications developed on its basis[3], have emerged as powerful tools to identify and study RBPs. RIC is based on the irradiation of living cells with ultraviolet (UV) light to generate covalent bonds between RNA and proteins that are in direct contact with each other. Subsequently, cells are lysed under denaturing conditions, and polyadenylated RNAs are isolated using deoxythymidine oligonucleotides (oligo(dT))-coupled beads. After extensive washes, the crosslinked proteins are eluted and identified by quantitative mass spectrometry[1,2]. RBP hits are defined as those proteins that are enriched in irradiated samples compared to controls that have not been irradiated but otherwise treated identically.

RIC has been applied to different eukaryotic systems, including mammalian cell lines, yeast, *Drosophila* embryos, and plant seedlings[1,2,4–7], successfully recovering many well-known RBPs. In addition, RIC has led to the identification of hundreds of new RBPs that were not previously related to RNA biology and that lack known RNA-binding domains (RBD)[4]. Nevertheless, it has been reported that RIC proteomes can be contaminated with DNA-binding and other proteins that could falsely be assigned as RBPs[3].

To unravel RBP function, we need to understand how RBPs respond to environmental and pharmacological cues. While RIC has successfully been employed toward this aim[7–9], experimental variability and technical noise limit its utility for such a challenging application.

Here we describe enhanced RIC (eRIC), which combines the use of a locked nucleic acid (LNA)-modified capture probe and more stringent capture and washing conditions. The performance of the eRIC and RIC protocols was directly compared using Jurkat cells. eRIC markedly increases capture specificity, reduces material requirement, and improves signal-to-noise ratios compared to precedent techniques. eRIC is particularly suitable for the detection of unconventional RBPs and the identification of dynamic changes among the RNA-bound proteome.

## Results

**Polyadenylic acid (poly(A)) tail-mediated capture of RBPs: considerations.** With suitable tools for de novo RBP discovery now in hand, we wanted to build on the principle of RIC and develop a method that is highly performant in comparative studies. To reach this goal, we aimed to reduce DNA and ribosomal RNA (rRNA) contaminations that could contribute inadvertently co-purifying proteins and increase background RBPs.

To minimize protein–protein interactions that resist the denaturing capture conditions, we preincubated cell lysates at 60 °C for 10–15 min[7], followed by centrifugation to remove insoluble material (Fig. 1a). By using a modified probe with LNA, which positions oligonucleotides optimally for hybridization with RNA, we profoundly increased the melting temperature between

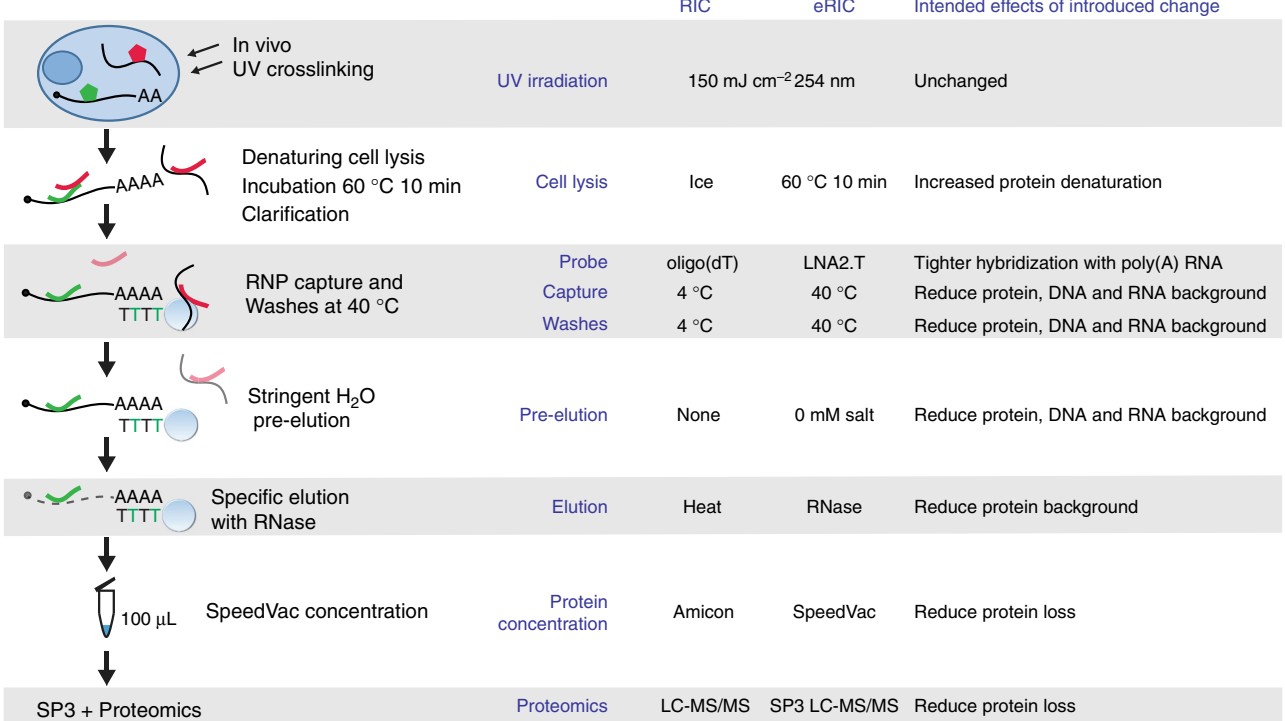

| | | RIC | eRIC | Intended effects of introduced change |
|---|---|---|---|---|
| In vivo UV crosslinking | UV irradiation | 150 mJ cm⁻² 254 nm | | Unchanged |
| Denaturing cell lysis Incubation 60 °C 10 min Clarification | Cell lysis | Ice | 60 °C 10 min | Increased protein denaturation |
| RNP capture and Washes at 40 °C | Probe | oligo(dT) | LNA2.T | Tighter hybridization with poly(A) RNA |
| | Capture | 4 °C | 40 °C | Reduce protein, DNA and RNA background |
| | Washes | 4 °C | 40 °C | Reduce protein, DNA and RNA background |
| Stringent H₂O pre-elution | Pre-elution | None | 0 mM salt | Reduce protein, DNA and RNA background |
| Specific elution with RNase | Elution | Heat | RNase | Reduce protein background |
| SpeedVac concentration | Protein concentration | Amicon | SpeedVac | Reduce protein loss |
| SP3 + Proteomics | Proteomics | LC-MS/MS | SP3 LC-MS/MS | Reduce protein loss |

**Fig. 1** Schematic representation of the eRIC method. RBPs are crosslinked to RNAs in vivo by irradiating cells with 254 nm UV light. Crosslinked proteins are isolated under denaturing conditions employing a LNA-modified probe and stringently washed under high temperature conditions using high salt concentrations first (to eliminate contaminants primarily based on hydrophilic protein–protein interactions) and low salt concentrations afterwards (to eliminate contaminants based on nucleic acid base pairing). Captured proteins are then eluted with RNase, concentrated to 100 μL to apply the Single-Pot Solid-Phase-enhanced Sample Preparation protocol (SP3, see main text: Analysis of RBPs identified by RIC versus eRIC), and identified by mass spectrometry. A comparison with the previous RIC protocol is included on the right. Green pentagon/line: native/denatured RBP; in red: contaminant protein. Black line with stretch of As: poly(A) RNA. Black line: contaminating non-poly(A) RNA/DNA. Light blue circle with stretch of T: capture probe coupled to magnetic beads (black and green Ts symbolize the alternation of DNA and LNA nucleotides; note that capture probe has actually 20 bases). LC-MS/MS: liquid chromatography–tandem mass spectrometry

the capture probe and poly(A) tails (from ~44 °C to ~77 °C), permitting more stringent purification conditions. A 20-mer bearing an LNA-thymine at every other position (LNA2.T) had previously been shown to effectively capture messenger RNA (mRNA)[10]. The probe design also includes a flexible C6 linker and a primary amino group at the 5' end, used to couple the LNA oligo to carboxylated magnetic beads (see Methods for detailed procedure). With this probe, all steps of the protocol, including capture and all washes, could be executed at 37–40 °C instead of 4 °C (Fig. 1a). Since salt stabilizes RNA–RNA and RNA–DNA duplexes, and hence favors contaminating nucleic acid pull down, we exploited the stability of LNA2.T–poly(A) RNA duplexes and incorporated a pre-elution step with pure water at 40 °C (Fig. 1a). This step proved instrumental for the efficient elimination of contaminant nucleic acids, such as rRNA and genomic DNA, without interfering with poly(A) capture (see below).

In RIC, the RNA pull down and the washes are performed at 4 °C to avoid interference with the oligo(dT)/poly(A) hybridization. This is followed by temperature-mediated elution at 50–55 °C (Fig. 1a). The increase in elution temperature could cause the co-elution of contaminants, including proteins directly associated with the beads employed for the pull down. To improve the specificity of the elution and to decrease background contaminants, we substituted the heat elution by RNase treatment at 37 °C (Fig. 1a). This elution strategy is more specific for RNA-bound proteins and is executed at a lower temperature than the washes. We termed this approach, which involves the use of the LNA2.T probe, increased capture and wash temperatures, the pre-elution, and the specific RNase-based elution, eRIC. We directly compared the performance of RIC and eRIC using Jurkat cells, a cell type characterized by a large nuclear volume compared to a relatively modest cytoplasmic volume. All experiments were performed in parallel.

**Reduction of rRNA and genomic DNA contamination by eRIC.** We first compared the RNA capture characteristics of the two protocols. Although the eRIC elution per se is RNase-mediated, an aliquot of purified eRIC material was heat-eluted to allow RNA analyses. Aliquots of the RIC and eRIC heat eluates were assessed in a bioanalyzer or reversely transcribed and subjected to quantitative polymerase chain reaction (qPCR) using intron-sensitive primers that amplify complementary DNA (cDNA) but not genomic DNA. Pull down of poly(A) RNA by eRIC is specific and capture probe-mediated, because no RNA is detected when uncoupled beads are used (Fig. 2b). Compared to RIC, eRIC exhibits a profoundly different RNA elution profile (Fig. 2a, b). While about 30% of the total RNA eluted by RIC corresponds to rRNA, it is only around 3% in the eRIC samples. Indeed, the bioanalyzer pattern for eRIC is dominated by an evenly distributed smear between ~500 and 4000 nucleotides that we attribute to polyadenylated RNAs, whereas RIC eluates predominantly show the rRNA bands (Fig. 2a). Capture of rRNA is UV-independent, and, interestingly depletion of the *28S* rRNA is more drastic than of the *18S* rRNA (Fig. 1b, c). These data suggest that *18S* rRNA co-purifies with poly(A) RNA either by means of a sufficiently long poly(A) stretch that is bound by the capture probe, or by hybridization of the rRNA to complementary sequences within poly(A) RNA. In line with these considerations, efforts to further reduce *18S* rRNA contamination were met by decreased poly(A) RNA yields. The qPCR results show that the higher temperature used in the eRIC protocol is not associated with RNA degradation (Fig. 2b).

DNA contamination was then estimated by qPCR analysis for multiple genes on aliquots of RIC and eRIC eluates without prior reverse transcription. The results demonstrate that eRIC dramatically reduces the contamination with genomic DNA by 10–100-fold (Fig. 2b, lower panel). By contrast, DNA contamination in RIC can reach the level of the cDNA for genes with low expression levels, as indicated by the levels of genomic DNA and cDNA for *ZNF80* (Fig. 2b).

Overall these results highlight that eRIC leads to a profound reduction in rRNA and genomic DNA contamination without compromising and possibly enhancing the capture of poly(A) RNA. Since the RNA analyses required heat rather than RNase elution of the eRIC samples for obvious reasons, the purity of eRIC samples may be even higher following the RNase-based elution of the original eRIC protocol.

**Analysis of RBPs identified by RIC versus eRIC.** We then subjected the proteins eluted according to the two protocols to downstream analysis. Following eRIC, the eluates are vacuum-concentrated using a SpeedVac, instead of the Amicon filters that RIC employs and that are commonly associated with protein loss and size bias. To exclude technical bias, SpeedVac-mediated concentration was also applied to RIC samples (see below). Sodium dodecyl sulfate-polyacrylamide gel electrophoresis (SDS-PAGE) and silver staining of eluted proteins shows patterns that differ profoundly from the input samples and that are absent from the non-irradiated controls, indicating the enrichment of specific RBPs after UV crosslinking by both RIC and eRIC (Fig. 2c). The band pattern of the eRIC and RIC eluates differs substantially (Fig. 2c), including proteins captured more efficiently by eRIC compared to RIC and vice versa. We also noticed that heat (re-)elution after the RNase-based elution of the eRIC samples yielded some proteins with similar migration as proteins eluted from the RIC samples (Fig. 2c), suggesting that these proteins are not RNA-bound. Therefore, RNase-mediated elution appears to be more specific for bona fide RBPs.

Specific enrichment for the known RBPs UnR, Nono, and HuR was confirmed in both the RIC and the eRIC samples by western blotting. Pull down of the RBPs by eRIC was at least equally efficient than for RIC (Fig. 2d).

To minimize pre-analytical sample loss, we introduced the highly sensitive Single-Pot Solid-Phase-enhanced Sample Preparation (SP3) protocol[11]. SP3 maximizes recovery of peptides for mass spectrometry and is compatible with the use of detergent throughout the procedure until the final wash. To facilitate the comparison between all other aspects of the RIC and eRIC protocols, and as described for the concentration step, SP3 was applied to both eRIC and RIC samples. Consequently, we used equal cell numbers for both methods, focusing on the differences in ribonucleoprotein (RNP) capture per se.

Protein eluates from two independent biological experiments each following the two different protocols were labeled by 10-plex tandem mass tag (TMT) and subjected to liquid chromatography–tandem mass spectrometry (LC-MS/MS) (Fig. 3a). Proteins that were significantly enriched in the crosslinked sample compared to the −UV control (false discovery rate (FDR) 0.05 (moderated $t$ test) and fold change (FC) > 2) were considered as "hits." Applying this criterion, we identified 683 and 588 RBPs in eRIC and RIC samples, respectively (Fig. 3b, d–e and Supplementary Data 1). We had anticipated that the more stringent eRIC purification procedure might reduce the number of identified RBPs, but the contrary was observed. Detailed data analysis reveals that the unique eRIC hits were also detected in the RIC samples, but that higher background levels in the −UV controls precluded their enrichment in +UV RIC samples and excluded them as statistically significant hits (Fig. 3c). When we compared the intensities of irradiated and non-irradiated samples of the 97 hits unique to eRIC, the normalized signal sum of these 97 proteins was significantly lower in the −UV control of eRIC in relation to

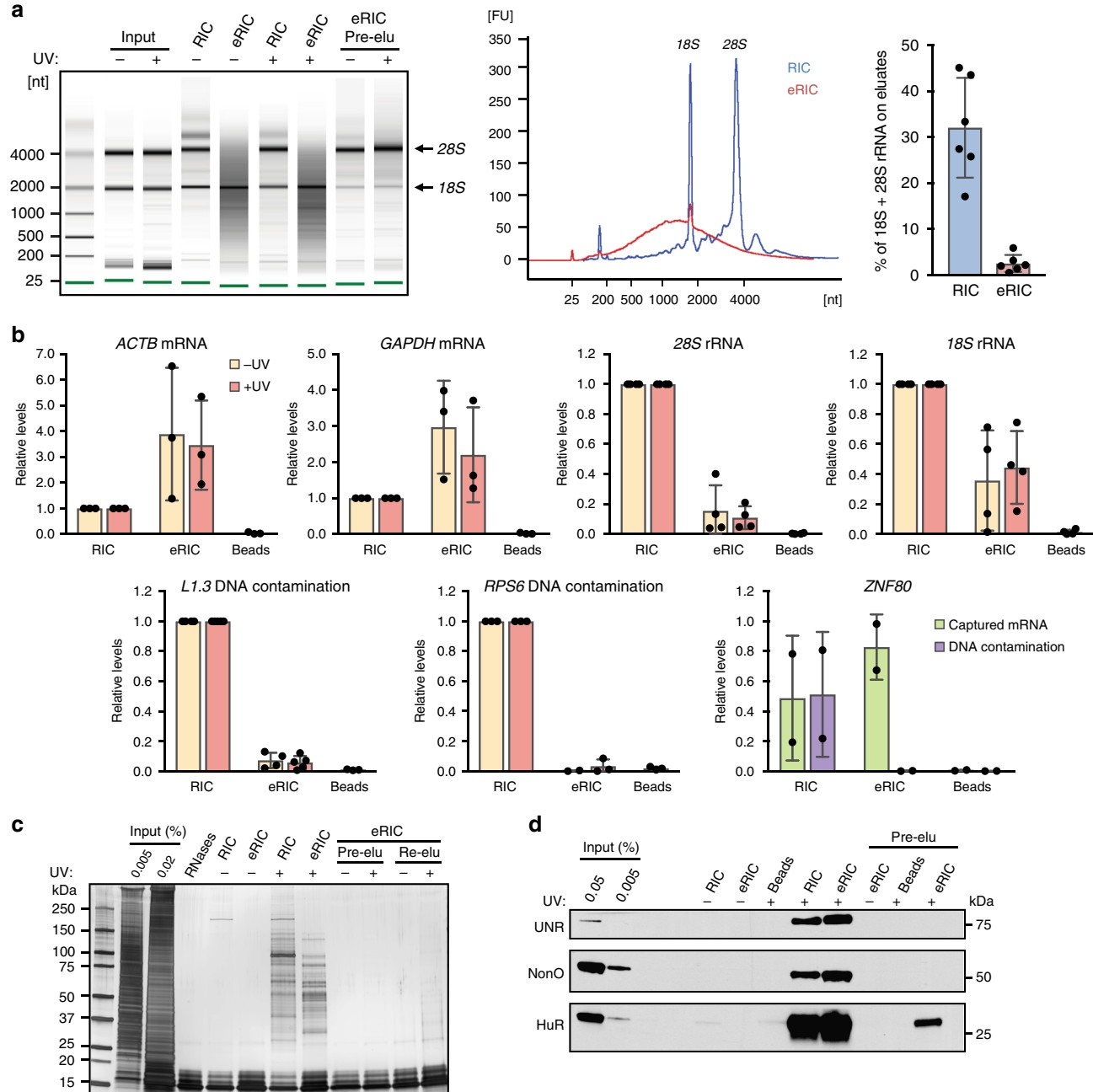

**Fig. 2** eRIC captures polyadenylated RNA and covalently crosslinked proteins with high specificity. **a**, **b** Nucleic acids isolated by eRIC, RIC, or using uncoupled beads along the eRIC protocol on irradiated material ("Beads") were analyzed using a 6000 Pico bioanalyzer (**a**) or by RT-qPCR (**b**). Note that the pre-elution step ("Pre-elu") incorporated in eRIC leads to an effective removal of rRNAs, specially the *28S* rRNA, without compromising the capture of poly(A) RNAs. **a** Representative result are shown in the left and middle panels; data on the right panel correspond to mean and standard deviation (s.d.) from six biologically independent experiments. −UV non-irradiated controls, +UV irradiated samples, [nt] length of RNA in number of nucleotides, [FU] fluorescence units. **b** Data correspond to mean and s.d. from at least three biologically independent experiments except for ZNF80 (two experiments). **c**, **d** Isolated proteins were separated by SDS-PAGE and silver-stained (**c**) or analyzed by western blot with antibodies to the positive control RBPs UNR, NonO, and HuR (**d**). The lane "RNases" contains a dilution of RNases A and T1. Note that lower bands in silver staining correspond to these enzymes. Re-elu heat elution performed after RNAse treatment. Note that a minor fraction of HuR appears to be associated with non-polyadenylated RNAs[39]

RIC, while the opposite was observed in the irradiated samples (Fig. 3f). Thus eRIC yields a dual benefit, reducing background and enhancing the specific pull down after crosslinking.

We then investigated the ontology of the RIC and eRIC hits. While classical RBPs (as defined by Gerstberger et al.[12]) were similarly identified by the two approaches (Fig. 3g), eRIC recovers more unorthodox RBPs, including enzyme and especially metabolic enzyme RBPs[4,5] (Fig. 3g). This enrichment is

particularly striking for enzymes of carbon metabolism, including the glycolytic pathway and the tricarboxylic acid cycle (Supplementary Fig. 1a), and enzymes involved in lipid, estrogen, and inosine 5'-phosphate metabolism (Supplementary Fig. 1b). The RNA-binding activity of at least some of these enzymes has previously been validated[4,13].

To evaluate whether eRIC detected new RBPs, we compared the list of eRIC hits with those of previous RIC experiments

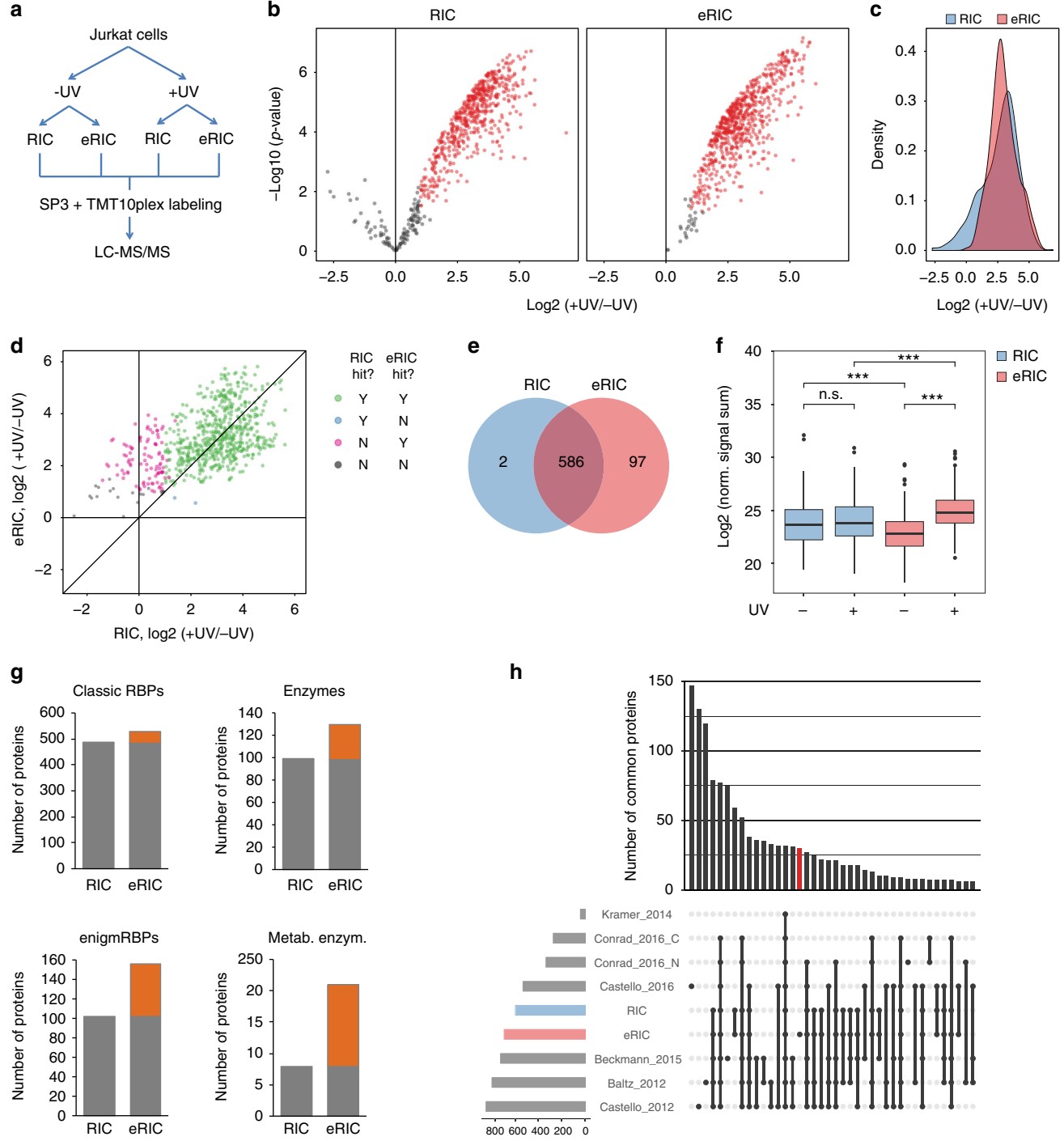

**Fig. 3** Superior performance of eRIC in RBP detection. **a** Scheme of the workflow of the direct comparative analysis of eRIC and RIC. Proliferating Jurkat cells were irradiated or not, lysed, and lysates equally split for eRIC or RIC analyses. Concentrated eluates were subjected to SP3, TMT-labeled, and analyzed by MS. **b** Volcano plots displaying the log2-fold change (FC) in irradiated (+UV) over non-irradiated (−UV) samples (x axis) and the p values (y axis) of the proteins identified by eRIC (right) and RIC (left). Proteins with FDR < 0.05 (moderated t test) and FC ≥ 2 were considered significantly enriched and are depicted in red. **c** Density of log2-FC between irradiated and non-irradiated samples of proteins identified by eRIC (red) and RIC (blue). Note the lack of enrichment over background of many proteins in RIC but not eRIC samples (leading left area of the blue curve). **d** Scatter plot comparing the averaged log2-FC in irradiated over non-irradiated samples of proteins detected by eRIC (y axis) and RIC (x axis). Hits recovered by both eRIC and RIC are displayed in green, hits unique to eRIC or RIC in magenta and blue, respectively, and proteins identified as background by both methods are shown in black. **e** Venn diagram comparing the number of hits identified by eRIC and RIC. **f** Normalized signal sum in irradiated and non-irradiated samples of the 97 hits exclusively identified by eRIC. ***p < 0.001 (Wilcoxon signed-rank test); n.s.: not significant. Center lines represent medians, box borders represent the interquartile range (IQR), and whiskers extend to ±1.5× the IQR; outliers are shown as black dots. **g** Number of known RBPs, enzymes, enigmRBPs[4], and metabolic enzymes identified by eRIC and RIC. Orange boxes represent the hits exclusive to eRIC. **h** UpSet plot showing the number of common proteins (i.e., intersections) between the eRIC and RIC experiments presented here or in previously published RBP datasets. Conrad_2016_C and _N refers to cytoplasmic and nuclear datasets[3], respectively. Data correspond to two biologically independent experiments

conducted in human cells[1–4,14,15]. eRIC yielded 30 candidate RBPs that were not detected either in the previous experiments nor in the RIC dataset from the Jurkat cells analyzed here (Fig. 3h). Overall, we identified 144 hits (Fig. 4 and Supplementary Data 1) that differ significantly between eRIC and RIC (comparison of enrichment over −UV controls, FDR 0.05 (moderated $t$ test) and FC > 2). Unsupervised clustering and posterior Gene Ontology (GO) analysis of these proteins revealed a high enrichment for terms associated with mRNA, such as "mRNA-processing," "mRNA splicing," and "mRNA transport" among the proteins preferentially recovered by eRIC (Fig. 4a). In contrast, RIC enriches for proteins that are mostly associated with rRNA-related terms, such as "ribosome biogenesis" and "rRNA processing" (Fig. 4a). Representative examples of RBPs differentially captured by eRIC and RIC are shown in Fig. 4b. Thus the pattern of RBPs recovered by eRIC and RIC reflects the nature of the RNAs captured with each method (Fig. 2).

**eRIC improves the detection of biological responses of RBPs**. A major motivation for the development of eRIC was the need for an optimized method to detect dynamic biological responses of the RNA-binding proteome to different experimental conditions, and we chose to evaluate the response of Jurkat cells to the

α-ketoglutarate antagonist dimethyloxalylglycine (DMOG) as a test case, because α-ketoglutarate is required as a co-factor by RNA demethylases[16,17] and we were curious to explore DMOG-induced changes in the RNA-bound proteomes. To reduce the influence of secondary effects, we incubated proliferating Jurkat cells with a modest concentration (0.5 mM) of DMOG for only 6 h. After crosslinking and lysis, we compared eRIC and RIC using two complete sets of biological replicates (Fig. 5a). eRIC led to the identification of 716 and 710 RBPs in dimethyl sulfoxide (DMSO; vehicle)- and DMOG-treated cells, respectively, while 673 and 662 RBPs where identified under identical treatment conditions by RIC (Supplementary Data 2), confirming the enhanced detection of RBPs by eRIC.

DMOG-responsive RBPs were defined in samples from UV-treated cells at an FDR of 0.05 (moderated $t$ test) and a consistent FC of at least 10% in each replicate. eRIC recovered a specific group of 20 RBPs with increased and of 41 RBPs with decreased RNA-binding after DMOG treatment (Fig. 5b and Supplementary Table 1), compared to 13 and 24 responsive RBPs, respectively, identified by RIC (Fig. 5b). The hits obtained by the two protocols display striking differences and only modest overlap (Fig. 5c and Supplementary Data 2). To better understand these differences, we compared the distribution of the differential ion intensities in eRIC and RIC eluates. These analyses showed reduced signal

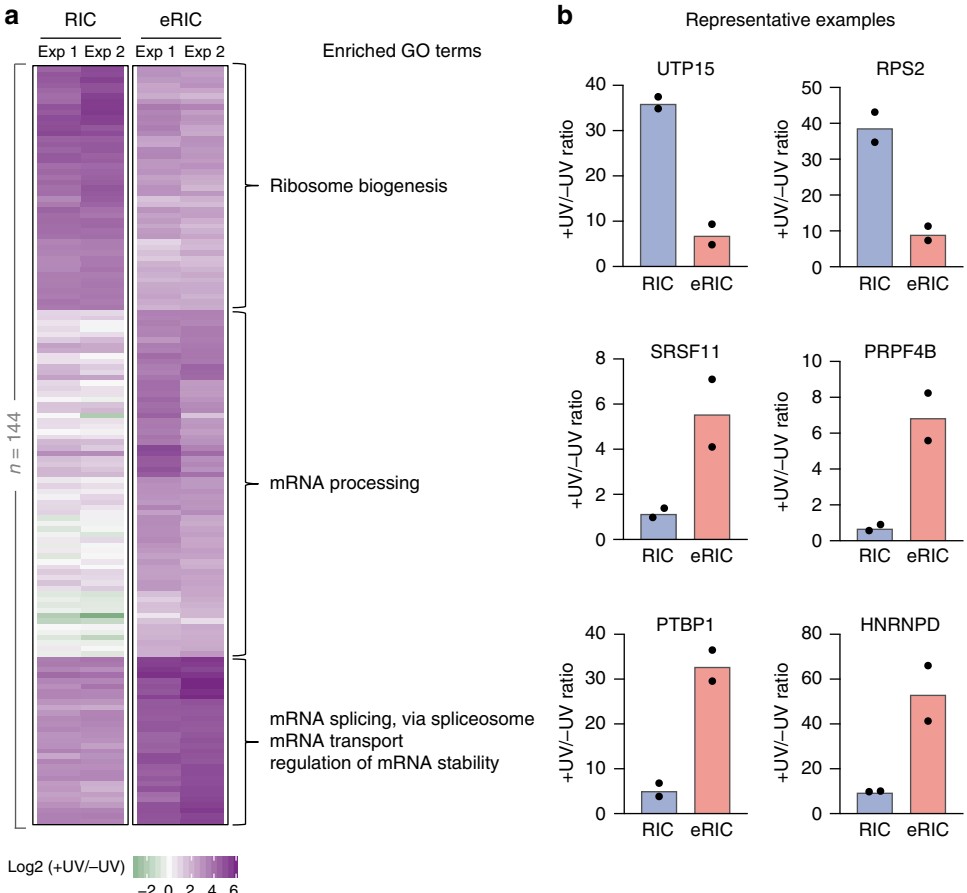

**Fig. 4** Differential RBP enrichment with eRIC. **a** Unsupervised clustering and GO analysis of proteins whose enrichments in irradiated (+UV) over non-irradiated (−UV) samples differ significantly between eRIC and RIC (FDR < 0.05 (moderated $t$ test) and FC > 2, 144 proteins). Three main clusters are observed that comprise RBPs preferentially recovered by RIC (top) or eRIC (middle and bottom). Note that proteins in the middle and bottom clusters markedly differ in their enrichment over background in RIC samples. Biological process GO terms enriched for each cluster are shown. Upper: "ribosome biogenesis" (−log10 ($p$ value) = 39.95; enrichment = 40.6), middle: "mRNA processing" (18.36; 14.85), bottom: "mRNA splicing via spliceosome" (33.48; 52.4), "mRNA transport" (8.52; 30.85) and "regulation of mRNA stability" (4.04; 17.08) (Fisher's Exact with FDR multiple test correction). **b** Fold change in irradiated over non-irradiated samples of representative example RBPs captured by eRIC and RIC. Data correspond to two biologically independent experiments (**a**, **b**); in **b**, data are shown as mean

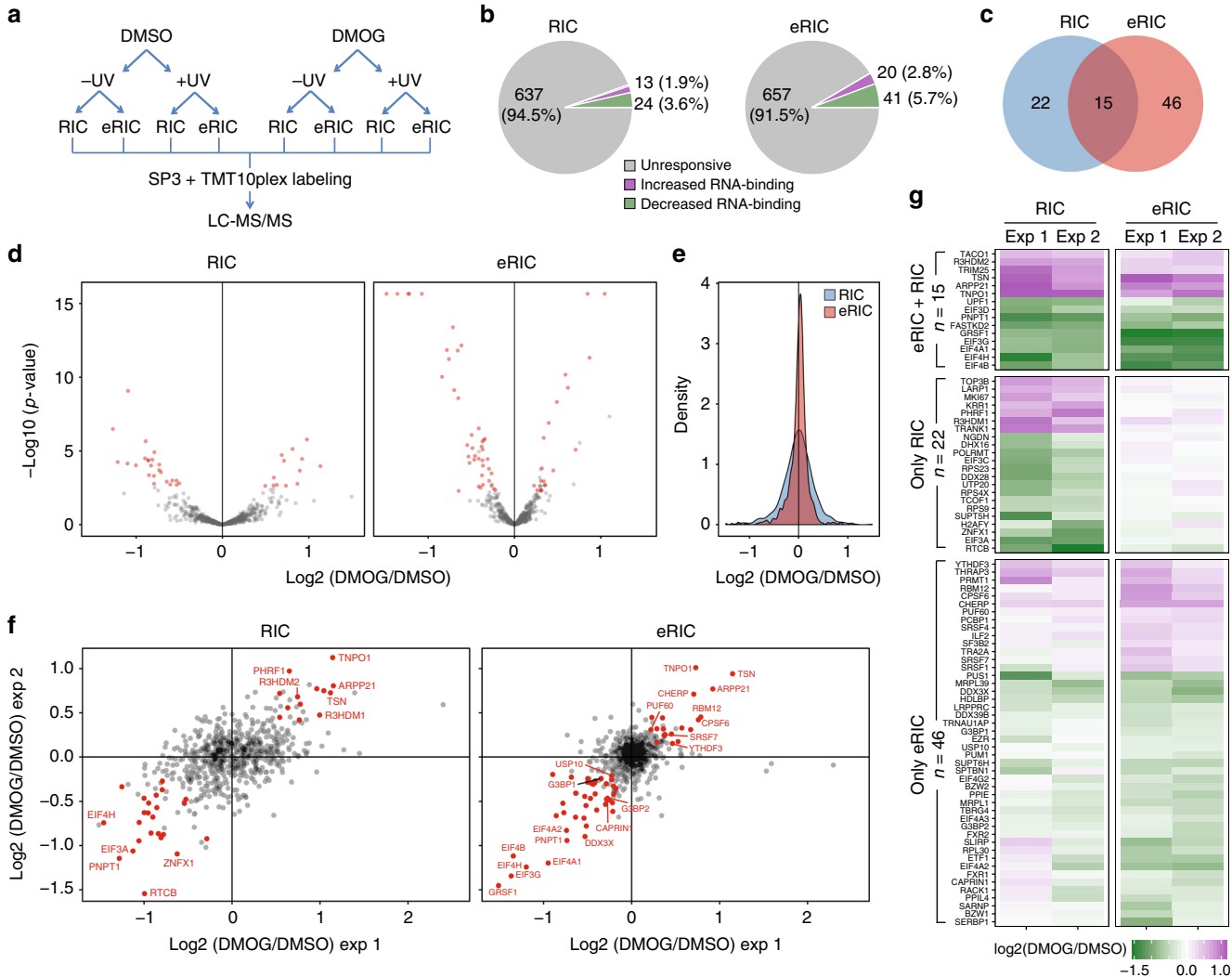

**Fig. 5** Superior performance of eRIC in comparative analyses of RBP responses. **a** Experimental design: Jurkat cells were incubated for 6 h with 0.5 mM DMOG or the vehicle DMSO. After irradiation, cells were lysed and lysates equally split for eRIC or RIC analyses. $n = 2$ independent experiments. **b** Pie charts summarizing the response of the RNA-bound proteomes to DMOG treatment identified by eRIC (right) or RIC (left). The number and percentage of proteins displaying constant (gray), increased (green), or decreased (violet) RNA association upon DMOG is shown. **c** Venn diagram comparing the number of DMOG-responsive RBPs identified by each method. **d** Volcano plots displaying the $p$ values ($y$ axis) and the log2-fold change (FC) in DMOG- versus DMSO -treated and irradiated samples ($x$ axis) of the proteins detected by eRIC (right) and RIC (left). Proteins with FDR < 0.05 (moderated $t$ test) and consistent FC of at least 10% in each replicate were considered as hits and are depicted in red. **e** Density of log2-FC in DMOG- over vehicle-treated and irradiated samples of the proteins identified by eRIC (red) and RIC (blue). Note the lower signal dispersion of the RBPs recovered by eRIC. **f** Scatter plots of detected proteins comparing the log2 ratios (DMOG/vehicle) of two independent experiments (exp 1/2). **g** Heat map showing the protein log2 ratios (DMOG/vehicle) of eRIC/RIC hits. Hits were divided according to their occurrence in eRIC and/or RIC and clustered. Upper: eRIC/RIC common hits, middle and bottom: RIC and eRIC exclusive hits, respectively

scatter and higher experimental reproducibility for the RBPs captured by eRIC (Fig. 5d–f), which increased the detection sensitivity for changes of the RNA-bound proteome.

For the 22 differential hits that were only recovered by RIC (Fig. 5c), we were struck by their complete lack of response in the eRIC samples (Fig. 5g and Supplementary Data 2). GO analysis revealed enrichment for rRNA-related terms and for constituents of the ribosome, pre-ribosome, and nucleolus (Fig. 6a), suggesting that they do not directly bind to poly(A) RNA and potentially co-purify with the contaminating rRNA (Fig. 2). GO analysis of DMOG-responsive hits shared between RIC and eRIC showed association with mRNA translation, especially for eukaryotic initiation factor 3 (eIF3) and eIF4 (Fig. 6a, b). By contrast, eRIC-specific DMOG-responsive hits were enriched for mRNA-

related functions such as mRNA transport and mRNA splicing or belong to complexes/structures that regulate mRNA metabolism, such as the spliceosomal complex and stress granules (Fig. 6a, b).

Taken together, eRIC facilitates sensitive comparative analyses of changes of the mRNA-bound proteome that escape RIC.

**eRIC implicates inhibition of the mammalian target of rapamycin (mTOR) pathway by DMOG.** DMOG treatment profoundly diminished the RNA binding of several translation initiation factors, especially of eIF3 and eIF4 (Fig. 6b and Supplementary Fig. 2a). Since DMOG has been reported to negatively affect the activity of the mTOR kinase[18], which phosphorylates the inhibitory protein 4EBP, we examined western blots for 4EBP phosphorylation. Indeed, following treatment with 0.5 mM

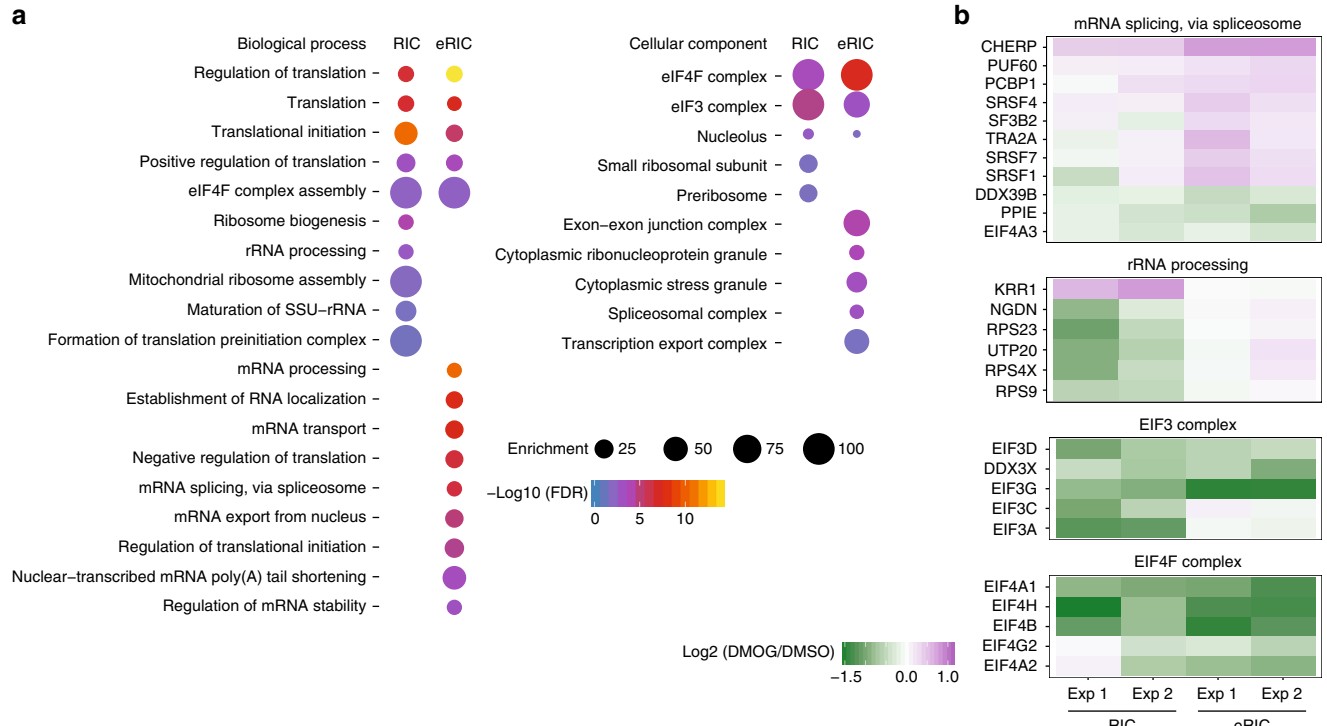

**Fig. 6** Improved detection of poly(A)RBP responses by eRIC. **a** Representative biological processes and cellular components enriched among the DMOG-responsive RBPs identified by eRIC and/or RIC (Fisher's Exact with FDR multiple test correction). **b** Examples of protein complexes or functionally related proteins that respond to DMOG with statistical significance

DMOG for 6 h, phosphorylation of 4EBP is severely reduced (Supplementary Fig. 2b). In addition, we observe reduced phosphorylation of the mTOR targets S6K and ULK1, as well as of the serine 2448 of TOR itself (associated with TOR activation) (Supplementary Fig. 2b). These results indicate that DMOG inhibits mTOR and thus activates 4EBP, inhibiting translation initiation and explaining the reduced RNA binding of eIF3 and eIF4. These findings exemplify the value of the eRIC data to shed light on a biological process.

**eRIC identifies N6-methyladenosine (m6A)-responsive RBPs that escape RIC.** RNA demethylases are α-ketoglutarate-dependent dioxygenases that can be inhibited by DMOG[19]. Such inhibition would be expected to increase the steady-state levels of m6A. We tested this possibility using dot blot assays on poly(A) RNA purified from Jurkat cells treated with 0.5 mM DMOG (or DMSO) for 6 h. This analysis indicated that DMOG incubation indeed increased the m6A modification of poly(A) RNA (Fig. 7a). We were thus curious to see whether RBPs associated with m6A biology also responded to the DMOG treatment, especially since m6A was recently shown to affect the RNA binding of different RBPs[20,21]. Strikingly, m6A-sensitive RBPs are significantly enriched among the eRIC hits (Fisher's exact test, p value = 0.00016). Of the 61 DMOG-regulated RBPs that we identified by eRIC, at least 18 (30%) were previously shown to be affected by m6A[20,21] (Fig. 7b and Supplementary Table 1). By contrast, only two of these were found by RIC (Fig. 7b). The direction of DMOG-induced changes corresponds well with predictions based on previous reports[20,21] (numbers in brackets in Fig. 7b). Representative examples of previously reported m6A readers (YTHDF3, CPSF6, PUF60, SRSF7) and m6A-repelled proteins (CAPRIN1, HDLBP, EIF4A1, G3BP2), as well as of proteins expected to be insensitive to m6A, are shown in Fig. 7c.

These results show that eRIC identifies changes of the poly(A) RNA-bound proteome concordant with the increase in steady-state m6A levels that are missed by RIC, further supporting the superior performance of eRIC in comparative studies.

## Discussion

RBPs are central to every aspect of gene regulation and many other processes in cell biology[5,22]. We describe a method developed on the basis of existing approaches, which fills a critical void in the available repertoire: a robust, highly reproducible, and sensitive method to detect biologically relevant changes in the RNA binding of RBPs in response to biological cues.

During the past few years, different methods have been developed to uncover and study the proteome-wide repertoire of RBPs[5]. Especially methods applied to living cells and organisms based on covalent RBP–RNA crosslinking have received much attention[1–3,23,24]. Taken together, these methods have revealed that the biological repertoire of RBPs is at least twice as large as previously anticipated, posing challenges and yielding opportunities for future discovery.

RIC derived its popularity from the stringency of purification enabled by the poly(A) tail–oligo(dT) hybridization. However, this method is by definition limited to RBPs bound to poly-adenylated RNAs. To address this, new approaches that also capture non-polyadenylated RNAs have been developed[23–25]. These strategies involve the in vivo incorporation of nucleotide analogs (such as 4-thiouridine or 5-ethynyluridine) into cellular RNA. However, metabolic labeling of RNA can inhibit rRNA synthesis and cause nucleolar stress[26], which is expected to cause significant secondary effects. Furthermore, the preferential labeling of newly synthesized RNAs, especially for those with high turnover rates, can bias RBP-binding analyses. Most recently, the discovery that silica-based matrices not only retain nucleic acids but also proteins that are covalently bound to these has added a

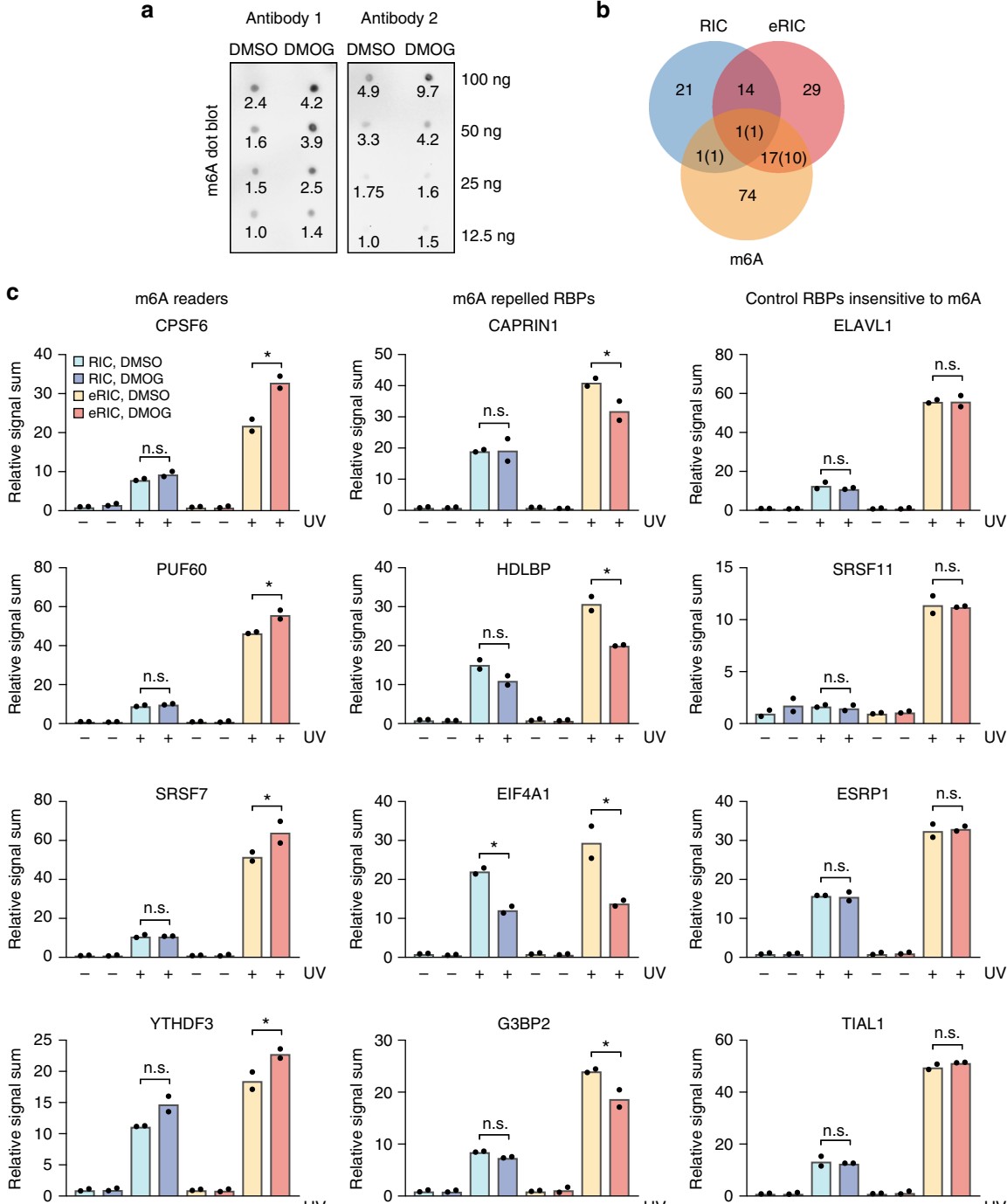

**Fig. 7** eRIC identifies m6A-responsive RBPs in vivo. **a** m6A dot blot of Jurkat cells treated with 0.5 mM DMOG or vehicle for 6 h, using two independent antibodies (Antibody 1: Abcam, Antibody 2: SySy). Serial dilutions of poly(A) RNA were blotted as indicated on the right. A quantification of the signal intensity based on image analysis is shown below each dot. Intensity is expressed relative to the lowest signal in the control. A representative blot of three biologically independent experiments is shown. **b** Overlap between DMOG-responsive RBPs identified by eRIC/RIC and m6A-regulated RBPs previously reported by Edupuganti et al.[21] or Arguello et al.[20]. Between brackets are the number of proteins with directions of DMOG-induced changes that coincide with previous reports. **c** Normalized signal sum in eRIC and RIC samples of representative examples of reported m6A readers (left), m6A-repelled RBPs (middle), and RBPs insensitive to m6A (right). −UV non-irradiated controls, +UV irradiated samples. eRIC and RIC values are expressed relative to the respective untreated control (−UV, DMSO). Data are shown as mean from two biologically independent experiments. Asterisk (*) indicates FDR < 0.05 (moderated $t$ test)

simple and powerful approach to the methodological repertoire for the discovery and isolation of RBPs, which is applicable to all classes of RNA[27]. However, this simple new method may offer limited selectivity especially against DNA-binding proteins, since silica matrices can bind both classes of nucleic acid[28].

In addition to the above considerations, the advantages of global RNA purification may be offset by disadvantages in the detection of poly(A) RNA-bound RBPs[23–25]. Poly(A) RNA represents only a small fraction of the total cellular RNA, and the absolute levels of many regulatory RBPs bound to this minority

fraction are also low. By contrast, RBPs bound to non-poly(A) RNA such as ribosomal proteins, rRNA processing factors, and ribosome-associated proteins represent a large fraction of MS/MS peptides and can mask the detection of poly(A) RNA-associated RBPs. For this reason, methods based on global RNA purification are expected to underperform in comparative studies aimed to identify responses of RBPs bound to poly(A) RNAs such as mRNAs and long non-coding RNAs.

Although RIC is principally limited to polyadenylated RNAs, problems with contamination by rRNA and DNA have been reported[1–3]. This problem is particularly evident with cells with a relatively small volume (e.g., lymphocytes and leukocytes that can be 20-times smaller than HeLa cells) (Fig. 2), where the content of genomic DNA relative to mRNA is shifted disfavorably for the study of RBPs. For reasons that are not entirely clear, rRNA contamination has also been noted in RIC studies[1–3].

Our data show that eRIC practically excludes DNA contamination even under challenging conditions and reduces rRNA contamination by at least one order of magnitude. These improvements, largely based on a further increase in the stringency of purification compared to RIC, shed an interesting light on many of the unorthodox RBPs that were identified in previous RIC studies. The observation that the more stringent purification protocol of eRIC yields even higher enrichments of these so-called enigmRBPs[4] (Fig. 3g and Supplementary Fig. 1) strongly supports their assignment as legitimate RBPs. This notion is consistent with the growing number of enigmRBPs for which RNA-related biological functions have been experimentally demonstrated[4,13,29,30].

We developed eRIC largely to provide a method for studies aimed to identify proteome-wide responses of RBPs to biological stimuli. Although RIC has been successfully used in this way[7–9], our comparative data clearly demonstrate the superior performance of eRIC. In DMOG-treated Jurkat cells, we found that eRIC yields a more consistent capture of RBPs within and between experiments (Fig. 5d–f), enhancing the sensitivity for the detection of significant changes by increasing the reproducibility (Fig. 5f) and statistical power (Fig. 5d). The lower signal scatter in eRIC experiments likely results from the elimination of contaminants.

Of the 721 proteins detected in our comparative experiments, 61 RBPs proved responsive to DMOG by eRIC, of which only 15 (less than one quarter) were identified by RIC (Fig. 5c). The 22 RIC-specific hits (Fig. 5c, g) largely correspond to non-poly(A) RNA binders (Fig. 6a, b), and thus were appropriately omitted by eRIC. We consider the depletion of contaminant rRNA/DNA and their associated proteins as one of the key elements explaining the reduced technical noise and overall improved performance of eRIC in comparative experiments, which allows the detection of biologically meaningful changes in mRNA–RBP interactions (see below).

Methylation of adenosine at the nitrogen-6 position (m6A) is one of the most abundant posttranscriptional modifications on mRNAs. The m6A modification of mRNAs responds to environmental cues and appears to be relevant in, e.g., stem cell differentiation, T cell function, and tumorigenesis[17,31]. The identity of the RBPs that are involved is still largely unknown. Two recent studies discovered RBPs with altered binding to m6A compared to unmodified RNA in vitro[20,21]. To the best of our knowledge, no equivalent proteome-wide studies have been performed on living cells. Remarkably, among the RBPs that eRIC identified to respond to DMOG treatment, we found a significant enrichment for proteins previously linked to m6A (Fisher's exact test, $p$ value = 0.00016; 18 of 61 RBPs) (Fig. 7b, c)[20,21]. Most of these (11 of 18) display the responses that would be expected based on prior knowledge of their modes of interaction with m6A (Fig. 7c).

Since some RBPs can be recruited or repelled by m6A in a context-dependent manner[21], one would expect less than complete concordance.

Having used DMOG treatment as a proof of concept for the use of eRIC for comparative analyses, this work provides a first in vivo analysis of the changes of the RNA-bound proteome in response to the inhibition of RNA demethylases and constitutes the first attempt to characterize the repertoire of m6A-regulated RBPs in vivo. Our datasets also implicate mTOR signaling in DMOG-induced effects on translation initiation factors and translation (Fig. 6a, b and Supplementary Fig. 2). Thus, while these findings are correlative, they demonstrate the utility and immediate practical relevance of eRIC for the discovery of posttranscriptional response pathways and mechanisms to biological or pharmacological cues. We expect that eRIC will also be broadly useful to study disease mechanisms (e.g., infection, malignant transformations, metabolic disorders) and the modes of action of therapeutic interventions.

## Methods

**Cell culture**. Jurkat cells (DSMZ, ACC-282) were maintained as a suspension culture in 175 cm$^2$ flasks (Falcon, 353028) in RPMI 1640 medium (Thermo Fisher Scientific, 21875034) supplemented with 10% heat-inactivated Fetal Bovine Serum (Gold, GE Healthcare) and penicillin/streptomycin (Sigma-Aldrich, P4333) in a humidified incubator at 37 °C and 5% CO$_2$.

**Coupling of the capture probe to beads**. The capture probe (HPLC purified; Exiqon) is composed of a primary amine at the 5' end, a flexible C6 linker, and 20 thymidine nucleotides in which every other nucleotide is a LNA: /5AmMC6/+TT +TT+TT+TT+TT+TT+TT+TT+TT+TT (+T: LNA thymidine, T: DNA thymidine)[10]. The probe was resuspended in nuclease-free water (Ambion) to a final concentration of 100 μM and coupled in DNA low binding tubes (Eppendorf) to carboxylated magnetic beads (Perkin Elmer, M-PVA C11) through the 5' amine as follows or kept at −20 °C until coupling. Bead slurry (50 mg/mL) was washed 3 times with 5 volumes of 50 mM 2-(N-morpholino)ethanesulfonic acid (MES) buffer pH 6. A 20 mg/mL solution of N-(3-dimethylaminopropyl)-N'-ethylcarbodiimide hydrochloride (EDC-HCl; Sigma-Aldrich) in MES buffer was freshly prepared. Five volumes were combined with 1 volume of 100 μM probe solution and this was added to pelleted washed beads originated from 1 volume of bead slurry (for one capture: 1.5 mL EDC solution + 300 μL probe solution + 300 μL bead slurry). Coupling was performed for 5 h at 50 °C and 800 rpm, with occasional pelleting. Beads were then washed twice with phosphate-buffered saline (PBS) and then incubated with 200 mM ethanolamine pH 8.5 for 1 h at 37 °C 800 rpm to inactivate any residual carboxyl residue. Coupled beads were finally washed three times with 1 M NaCl and stored in 0.1% PBS–Tween at 4 °C.

**Recycling of LNA2.T-coated beads**. Coupled beads can be reused several times. To do so, the poly(A) stretches interacting with the LNA probe, expected to be resistant to the RNA digestion, have to be eluted by other mean. Also, any trace of RNases from the elution has to be eliminated. Beads were reused a few times in this work and at least eight times in other works, with optimal results.

Coupled beads employed for a capture (300 μL) were resuspended in 400 μL of nuclease-free water (Ambion), transferred to a 1.5 mL tube, and incubated for 5–10 min at 95 °C 800 rpm. Immediately afterwards, before bead slurry cool down, beads were collected by magnetic force, and the supernatant discarded. Beads were then washed 3 times with 5 volumes of water and 3 times with 5 volumes of lysis buffer and stored in 0.1% PBS–Tween at 4 °C until use.

**Cell lysis in eRIC/RIC**. In all, $1.0–1.3 \times 10^8$ proliferating Jurkat cells at a density of about $1–1.5 \times 10^6$ cells/mL were employed per sample. Where stated, 0.5 mM DMOG (Cayman Chemical Company, 71210) or an equivalent volume of DMSO (vehicle, Merck 1.02950.0500) was added to the culture medium for 6 h prior to processing. DMSO concentration in the medium was 0.023% v/v. Cells were collected by centrifugation at $400 \times g$ for 5 min at 4 °C, resuspended in 40 mL of cold PBS, and split into two 145 × 20 mm$^2$ petri dishes (Greiner Bio-One, 639102), which were deposited on a metal plate pre-cooled on ice and irradiated with 150 mJ/cm$^2$ at 254 nm UV light in a Spectrolinker XL-1500 (Spectronics Corporation). Irradiation was omitted in −UV controls. While constantly maintaining 4 °C, cells were transferred to 50 mL conical centrifuge tubes, pelleted at $400 \times g$ for 5 min, and lysed in 7.5–10 mL of ice-cold lysis buffer (see composition below) supplemented with cOmplete Protease Inhibitor Cocktail (Roche, 11873580001). To enhance homogenization, samples were passed 3–5 and 6–10 times through syringes with 22-Gauge (0.7 mm diameter) and 27-Gauge needles (0.4 mm diameter), respectively, snap frozen in liquid nitrogen, and kept at −80 °C for several days until further processing.

**Capture of RNP complexes using eRIC.** Cell lysates were thawed in a 37 °C water bath, incubated for 15 min at 60 °C, quickly cooled down on ice, and clarified 5 min at max speed and 4 °C. Dithiothreitol (DTT) extra (5 mM) was added to the samples. LNA2.T-coupled beads were equilibrated in lysis buffer (3 buffer exchanges with 3 volumes of lysis buffer each). After saving 100 μL as input, lysates were incubated with 300 μL of equilibrated LNA2.T-coupled beads for 1 h at 37–40 °C (inside an incubator) with gentle rotation to capture RNA–protein complexes. Beads were collected with a magnet, and the supernatant was transferred to a fresh tube for a second round of capture. Beads were subjected to successive rounds of washes, each of them performed for 5 min with gentle rotation at 37–40 °C (inside an incubator) with 10 mL of the corresponding buffer pre-warmed to 37–40 °C. We performed 1 wash with lysis buffer and 2 successive washes with each of the buffers 1, 2 and 3 (see composition below). Pre-elution was performed in 220 μL of nuclease-free water (Ambion) for 5 min at 40 °C and 800 rpm. Afterwards, the bead suspension was divided into two aliquots, one of 200 μL for the RNase-mediated elution for protein analysis and one of 20 μL that was heat-eluted for RNA/DNA analyses. For the RNase-mediated elution, beads were resuspended in 150 μL of 1× RNase buffer (see composition below), 5 mM DTT, 0.01% NP40, ~200 U RNase T1 (Sigma-Aldrich, R1003–100KU), and ~200 U RNase A (Sigma-Aldrich, R5503) and incubated at 37 °C 800 rpm for 30–60 min. Beads were then collected with a magnet, and the supernatant transferred to a fresh tube, which was placed again on a magnet (to fully remove any trace of beads) before saving the supernatant. Eluates were maintained on ice until finishing the second round of capture. Then combined eluates were supplemented with 2 μL 10% SDS and concentrated using a SpeedVac until reaching a volume of <100 μL (30–45 min at 37 °C), snap frozen, and stored at −80 °C. Heat elution was performed on the beads reserved ad hoc with 15 μL of elution buffer (see composition below) at 95 °C 800 rpm for 5 min. Beads were immediately collected, and the supernatant quickly recovered (before temperature drops). Any trace of beads was eliminated by a second round of collection as explained before.

Lysis buffer: 20 mM Tris-HCl (pH 7.5), 500 mM LiCl, 1 mM EDTA, 5 mM DTT, 0.5% (w/v) LiDS.

Buffer 1: 20 mM Tris-HCl (pH 7.5), 500 mM LiCl, 1 mM EDTA, 5 mM DTT, 0.1% (w/v) LiDS.

Buffer 2: 20 mM Tris-HCl (pH 7.5), 500 mM LiCl, 1 mM EDTA, 5 mM DTT, 0.02% (v/v) NP40.

Buffer 3: 20 mM Tris-HCl (pH 7.5), 200 mM LiCl, 1 mM EDTA, 5 mM DTT, 0.02% (v/v) NP40.

10× RNase buffer: 100 mM Tris-HCl(pH 7.5), 1.5 mM NaCl

Elution buffer: 20 mM Tris-HCl (pH 7.5), 1 mM EDTA.

**Capture of RNP complexes using RIC.** Lysates were thawed in a 37 °C water bath, and after taking 100 μL as input, they were incubated with 300 μL of equilibrated oligo(dT)$_{25}$ magnetic beads (NEB) at 4 °C for 1 h with gentle rotation. Beads were washed with 10 mL of ice-cold buffers. RNP complexes were eluted in 165 μL of elution buffer for 5 min at 55 °C and 800 rpm. An aliquot of 15 μL was taken and used for RNA/DNA analyses. The remaining 150 μL were combined with 10× RNase buffer, 1 M DTT, and 1% NP40 (final concentrations: 1× RNase buffer, 5 mM DTT, 0.01% NP40) and ~200 U RNase T1 and RNase A (Sigma-Aldrich). RNA was digested for 60 min at 37 °C. Two rounds of capture were performed and combined eluates were concentrated and stored as described for eRIC. Final volume of RIC eluates was adjusted to the volumes of the corresponding eRIC eluates.

**Sample preparation for mass spectrometry and TMT labeling.** Captured proteins were reduced in 10 mM DTT in 50 mM HEPES pH 8.5 at 56 °C for 30 min and alkylated with 20 mM 2-chloroacetamide in 50 mM HEPES pH 8.5 for 30 min at room temperature in the dark. Samples were prepared using the SP3 protocol[11]. Proteins were digested by trypsin (Promega) at 37 °C overnight using an enzyme-to-protein ratio of 1:50. Peptides were labeled with TMT10plex Isobaric Label Reagent (Thermo Fisher Scientific) according to the manufacturer's instructions. For further sample clean up, an OASIS HLB μElution Plate (Waters) was used. Offline high pH reverse phase fractionation was carried out on an Agilent 1200 Infinity high-performance liquid chromatography system, equipped with a Gemini C18 column (3 μm, 110 Å, 100 × 1.0 mm$^2$, Phenomenex).

**Mass spectrometric data acquisition.** An UltiMate 3000 RSLC nano LC system (Dionex) fitted with a trapping cartridge (μ-Precolumn C18 PepMap 100, 5 μm, 300 μm i.d. × 5 mm, 100 Å) and an analytical column (nanoEase™ M/Z HSS T3 column 75 μm × 250 mm C18, 1.8 μm, 100 Å, Waters) was employed. Trapping was carried out with a constant flow of solvent A (0.1% formic acid in water) at 30 μL/min onto the trapping column for 6 min. Subsequently, peptides were eluted via the analytical column with a constant flow of 0.3 μL/min with increasing percentage of solvent B (0.1% formic acid in acetonitrile) from 2 to 4% in 4 min, from 4 to 8% in 2 min, from 8 to 28% in 96 min, and finally from 28 to 40% in 10 min. The outlet of the analytical column was coupled directly to a QExactive plus mass spectrometer (Thermo Fisher Scientific) using the proxeon nanoflow source in positive ion mode.

The peptides were introduced into the QExactive plus via a Pico-Tip Emitter 360 μm OD × 20 μm ID; 10 μm tip (New Objective) applying a spray voltage of 2.3 kV. The capillary temperature was set at 320 °C. Full mass scan was acquired with mass range 350–1400 m/z in profile mode in the FT with resolution of 70,000. The filling time was set at maximum of 100 ms with a limitation of 3 × 10$^6$ ions. Data-dependent acquisition was performed with the resolution of the Orbitrap set to 35,000, with a fill time of 120 ms and a limitation of 2 × 10$^5$ ions. A normalized collision energy of 32 was applied. The instrument was set to alternate between MS and data-dependent MS/MS-based acquisition with up to a maximum of 10 MS/MS events per cycle. A minimum AGC trigger of 2e2 and a dynamic exclusion time of 30 s were used. The peptide match algorithm was set to "preferred" and charge exclusion to "unassigned," charge states +1 and +5 to +8 were excluded. MS/MS data was acquired in profile mode.

**Mass spectrometric data analysis.** IsobarQuant[32] and Mascot (v2.2.07) were used to process the acquired data, which was searched against the Uniprot *Homo sapiens* proteome database UP000005640, which contains common contaminants and reversed sequences. The following modifications were included into the search parameters: Carbamidomethyl (C) and TMT10 (K) (fixed modifications), Acetyl (N-term), Oxidation (M), and TMT10 (N-term) (variable modifications). A mass error tolerance of 10 ppm and 0.02 Da was set for the full scan (MS1) and the MS/MS spectra, respectively. A maximum of two missed cleavages were allowed and the minimal peptide length was set to seven amino acids. At least two unique peptides were required for protein identification. The FDR on peptide and protein level was set to 0.01. The R programming language (ISBN 3–900051–07–0) was used to analyze the raw output data of IsobarQuant. Potential batch effects were removed using the limma package[33]. A variance stabilization normalization was applied to the raw data using the vsn package[34]. Individual normalization coefficients were estimated for crosslinked and non-crosslinked conditions. During the DMOG versus DMSO comparison, an additional blocking factor for the protocol (RIC or eRIC) was chosen. Normalized data were tested for differential expression using the limma package. The replicate factor was included into the linear model. For comparisons between crosslinked versus non-crosslinked, hits were defined as those proteins with an FDR <5 % and an FC > 2. In eRIC/RIC comparative experiments, proteins were first tested for their enrichment over −UV controls, and the intensity of the proteins enriched in at least one condition were compared in the corresponding +UV samples. The R package fdrtool[35] was employed to calculate FDRs using the *t* values from the limma output. Proteins with an FDR < 5% and a consistent FC of at least 10% in each replicate were defined as hits. The ggplot2 R package[36] was used to generate the graphical representations.

**Hit classification and GO analysis.** In the "single point" eRIC/RIC experiments, enzymes were defined as those proteins listed in the six enzyme commission groups (EC 1–6) in the IntEnz database (release May 2017). Metabolic enzymes were defined as those enzymes that map to Metabolism in Reactome, plus the subunits of the ATP synthase and the respiratory chain complexes. enigmRBPs are defined as those eRIC/RIC hits that do not have a known RNA-binding domain (based on Hentze et al.[5]) and that lack known RNA-binding functions (based on Gerstberger et al.[12]). Overlaps with other RIC data sets were displayed in UpSet plots[37].

Comparison of DMOG-responsive hits identified by eRIC and RIC with previously reported m6A-regulated RBPs[20,21] was conducted using Venny 2.1.0 (Oliveros, J.C. (2007–2015), http://bioinfogp.cnb.csic.es/tools/venny/index.html). Analysis were restricted to those proteins detected by the eRIC/RIC comparative experiments. Fisher's exact tests were used to calculate enrichment of m6A-responding proteins among eRIC and RIC samples.

GO-term enrichment analysis were performed with AmiGO 2 (powered by PANTHER), using the following parameters: analysis type: PANTHER overrepresentation test; reference list: *Homo sapiens* (all genes in database); annotation data set: GO biological process complete or GO cellular component complete, as indicated; test type: Fisher's exact with FDR multiple test correction. Overrepresented GO terms were manually curated, and only selected terms were included in the main figures due to space constrains. The full lists of GO-enriched terms are provided in Supplementary Data 3 and Supplementary Data 4. The ggplot2 R package[36] was used to generate the graphical representations.

**Bioanalyzer and real-time PCR.** The concentration of captured RNA (heat-eluted) was estimated using a NanoDrop spectrophotometer (Thermo Fisher Scientific). To determine the profile of captured RNA, 1 μL of each sample was diluted to 5–10 ng/μL and analyzed in an Agilent 2100 Bioanalyzer System using the RNA 6000 Pico Kit, following the manufacturer's indications. Where stated, total RNA from whole-cell lysates was purified using the Quick-RNA MicroPrep Kit (Zymo) and analyzed similarly.

In all, 2–5 μL of the undiluted captured RNA were reverse transcribed into cDNA using SuperScript II (Life Technologies) and random hexamers (Life Technologies), according to the manufacturer's instructions. Real-time qPCR was performed using SYBR Green PCR Master Mix (Life Technologies, 4309155) in a QuantStudio 6 Flex system (Life Technologies) with the following primers (all from

5′ to 3′, forward: f, reverse: r): 28 S rRNA (f: TTACCCTACTGATGATGTGTTG
TTG, r: CCTGCGGTTCCTCTCGTA), RPS6 (f: TGAAGTGGACGATGAACGCA,
r: CCATTCTTCACCCAGAGCGT), ZNF80 (f: CTGTGACCTGCAGCTCATCCT,
r: TAAGTTCTCTGACGTTGACTGATGTG). From ref. [2]: β-actin (f: CGCGAGA
AGATGACCCAGAT, r: TCACCGGAGTCCATCACGAT), GAPDH (f: GTGGAG
ATTGTTGCCATCAACGA, r: CCCATTCTCGGCCTTGACTGT) and 18 S rRNA
(f: GAAACTGCGAATGGCTCATTAAA, r: CACAGTTATCCAAGTGGGAGAG
G). From ref. [3]: L1.3 (f: TGAAAACCGGCACAAGACAG, r: CTGGCCAGAACT
TCCAACAC).

**Western blotting and silver staining**. Proteins co-purified by eRIC or RIC or
present in whole-cell lysates (inputs) were separated by SDS-PAGE and subjected
to silver staining following standard procedures or transferred onto a nitrocellulose
membrane and analyzed by western blotting. Primary antibodies against the
following proteins at the indicated dilutions were used: Cold shock domain-
containing protein E1 (CSDE1)/UNR (Proteintech, 13319–1-AP, 1:5000), Non-
POU domain-containing octamer-binding protein (NonO) (Novus Biologicals,
NBP1–95977, 1:1000), and ELAV-like protein 1 (ELAVL1)/Hu-antigen R (HuR)
(Proteintech, 11910–1-AP, 1:5000). As a secondary antibody, a Goat anti-rabbit
immunoglobulin G (IgG) horseradish peroxidase (HRP) (Santa Cruz Biotechnol-
ogy, sc-2030, 1:5000) was employed. Uncropped scans are provided in Supple-
mentary Fig. 3a.

Proliferating Jurkat cells at a density of about $1 \times 10^6$ cells/mL were
incubated with 0.5 mM DMOG (Cayman Chemical Company, 71210) or an
equivalent volume of DMSO (vehicle, Merck 1.02950.0500) for 6 h. Subsequently,
cells were pelleted ($400 \times g$ 5 min 4 °C), washed with ice-cold PBS, and lysed in
ice-cold RIPA buffer (50 mM Tris-HCl, pH 7.4, 1% NP-40, 0.5% Na-deoxycholate,
0.1% SDS, 150 mM NaCl, 2 mM EDTA, 50 mM NaF) supplemented with
proteinase inhibitors (Roche, 11873580001) and phosphatase inhibitors (Sigma-
Aldrich, 04906845001). After clarification, proteins were separated by SDS-PAGE
and transferred onto a nitrocellulose membrane. Primary antibodies to the
following proteins at the indicated dilutions were used: anti-GAPDH (Sigma-
Aldrich, G9545, 1:100,000), all the rest from Cell Signaling: phospho-4EBP1
(Ser65) (9451 S, 1:10,000), 4EBP1 (9644 S, 1:50,000), phospho-p70 S6 Kinase
(Thr389) (9205, 1:2000), p70 S6 Kinase (2708, 1:50,000), phospho-ULK1
(Ser757) (14202, 1:10,000), and ULK1 (8054, 1:5000), phospho-mTOR(Ser2448)
(5536 P, 1:5000). A Goat anti-rabbit IgG HRP (Santa Cruz Biotechnology, sc-2030,
1:5000) was used as secondary antibody. Uncropped scans are provided in
Supplementary Fig. 3b.

**m6A dot blotting**. Aliquots of the heat-eluted RNA of the same eRIC samples
analyzed by MS (−UV controls) were used to estimate m6A levels. RNA was
incubated for 10 min at 65 °C and immediately placed on ice. Concentration was
estimated with a NanoDrop spectrophotometer (Thermo Fisher Scientific) and
serial dilutions were prepared in order to obtain 100, 50, 25, and 12.5 ng/µL. One
microliter of each dilution was directly pipetted onto a Zeta Probe membrane (Bio-
Rad), air dried for ~10 min, and crosslinked twice with 120 mJ/cm² at 254 nm in a
Spectrolinker XL-1500 (Spectronics Corporation). The membrane was washed with
0.05% PBS–Tween (PBS-T) and blocked for 1 h with 5% skimmed milk in PBS-T.
Primary antibodies were incubated in blocking solution overnight at 4 °C, followed
by 3 rinses with PBS-T, secondary antibody incubation in blocking solution for 1 h
at room temperature, 3 washes with PBS-T, and development using ECL (Milli-
pore, WBKLS0500). Antibodies used were: anti-m6A (Abcam, ab151230, 1:2000
and Synaptic Systems, 202 003, 1:2000) and Goat anti-Rabbit IgG-HRP (Abcam,
1:20,000).

## Data availability

All data generated during this study are included in this published Article (and its
Supplementary Information files). The mass spectrometry proteomics data have
been deposited to the ProteomeXchange Consortium via the PRIDE[38] partner
repository with the dataset identifiers PXD010941 (eRIC and RIC of Jurkat cells)
and PXD010942 (eRIC and RIC of Jurkat cells exposed to 0.5 mM DMOG or
vehicle (DMSO) for 6 h).

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

## Acknowledgements

We thank Mikhail Savitski from the EMBL Proteomics Core Facility (Heidelberg, Germany) for generous expert support with proteomic analyses. We also thank all members of the Hentze laboratory, especially Rastislav Horos, for fruitful discussions.

## Author contributions

J.I.P.-P. and M.W.H. conceived the project and designed the experiments. J.I.P.-P. performed the experimental work with help from B.R., Y.Z., and A.B. M.R. and F.S. conducted the MS data acquisition and analyses, respectively, and T.S. performed the hit classifications. J.I.P.-P. and M.W.H. wrote the manuscript with input from all authors.

## Additional information

**Competing interests:** The authors declare no competing interests.

