## [Peer Review File · Nature Communications]

Reviewers' Comments:

Reviewer #1:

Remarks to the Author:

My apologies for the delay in my review. This is a very well written manuscript describing a great new addition to the methods for purifying RNA-protein complexes. The method is described well and the data are very convincing. Typically, I find it difficult to describe both a method and a biological story in one manuscript but the authors do a good job of this and neither story is neglected. The only criticism I have is that the figure legends are way too abridged and content is lost. Simple things like "what does -eRIC and +eRIC mean" are not described. This is essentially true for every figure panel. The legends should be much more comprehensive in detail given the amount of data being represented in every figure. The same is true for the supplemental data.

Reviewer #2:

Remarks to the Author:

Perez-Perri et al. describe an improved workflow for capturing the RNA interactome. The improvements are based on the previously published RIC (i.e. RNA-interactome capture) technique. RIC involves purification of poly(A)-mRNAs and bound proteins through oligo(dT)-coupled magnetic beads. The improved technique (eRIC, i.e. enhanced RIC) employs a modified probe which allows for more stringent washing steps and therefore reduces background proteins. These include mostly ribosomal RNA. The method is first compared with the existing RIC method using Jurkat cells and in a second experiment quantitative changes are studied in response to DMOG, which is affecting the RNA demethylases. The developed eRIC method could successfully be applied to monitor changes in the RBPome.

All experiments are well-designed and sufficiently demonstrate the improvements of eRIC. The manuscript is well-written and clearly describes the results. I do not have major comments. I suggest few very minor changes and corrections before publication of this manuscript at Nature communications.

- The figures are too full. I doubt that everything will be readable at final printing size. As the number of supplemental figures is very low I suggest to move some of the results to supplementary material and/or include more figures in the main text (the maximum number of figures allowed by Nature communications is not yet reached)
- Abstract, line 20: "signal-to noise ratios" change to "signal-to-noise ratios"
- Methods, line 324: "coming from 1 volume" sounds rather colloquial. I suggest to re-phrase
- Methods, lines 342, 346 and 483: "400g" change to "400 x g"
- Methods, line 395: "ON" change to "overnight" (not clear)
- Methods, line 403: "were employed" change to "was employed"

Point by point answer to REVIEWERS' COMMENTS.

We are thankful to both reviewers for their positive comments and useful suggestions.

Reviewer #1 (Remarks to the Author):

My apologies for the delay in my review. This is a very well written manuscript describing a great new addition to the methods for purifying RNA-protein complexes. The method is described well and the data are very convincing. Typically, I find it difficult to describe both a method and a biological story in one manuscript but the authors do a good job of this and neither story is neglected. The only criticism I have is that the figure legends are way too abridged and content is lost. Simple things like “what does -eRIC and +eRIC mean” are not described. This is essentially true for every figure panel. The legends should be much more comprehensive in detail given the amount of data being represented in every figure. The same is true for the supplemental data.

The only criticism I have is that the figure legends are way too abridged and content is lost. Simple things like “what does -eRIC and +eRIC mean” are not described. This is essentially true for every figure panel. The legends should be much more comprehensive in detail given the amount of data being represented in every figure.

In an attempt to accommodate the brief style of Nature journals, we admittedly overemphasized brevity at the expense of clarity. We carefully checked and improved all of the figure legends. We also explain the abbreviations, new terms and color codes, and define those that were not defined before. Finally, we introduced changes of some figures panels to make them more intuitive and clear:

A more comprehensive graphic representation and description of the method was included in Fig. 1. All graphical objects are properly defined now.

A definition of “-eRIC and +eRIC” was added (Fig. 2, 3, 4, 7, Supplementary Fig. 1 and Supplementary Fig. 2).

The lane “RNases” was defined for Fig. 1d.

An explanation of the scheme presented in Fig. 3a was included in the legend.

In the legends of Fig. 7 and Supplementary Table 1, the term “DMSO” was defined as the vehicle for clarity.

The orange boxes in Fig. 3g were defined.

A more intuitive title as well as a definition of the expression Conrad_2016_C and _N was introduced for Fig.3h.

An explanation of the difference between middle and bottom clusters in Fig.4a was introduced.

The clusters in Fig. 5g were named in the figure and defined in the legend. Furthermore, each of the columns was labeled.

The term “exp 1/2 was defined in the legend of Fig. 5f.

The seeding strategy and the definition of the numbers on the right side of Fig. 7a was included in the legend. A definition of the numbers below the dots was also included.

The same is true for the supplemental data.

We added information to the legends of the supplementary figures to increase their clarity. We also added a figure legend to the supplementary table 1 that did not have a legend in the originally submitted version. Descriptive legends were also added to the Supplementary Dataset 1 and 2 and the Supplementary Note 1 and 2. Finally, we incorporated a description of the content of each individual sheet of the Supplementary Dataset 1 and 2 (excel files).

Reviewer #2 (Remarks to the Author):

Perez-Perri et al. describe an improved workflow for capturing the RNA interactome. The improvements are based on the previously published RIC (i.e. RNA-interactome capture) technique. RIC involves purification of poly(A)-mRNAs and bound proteins through oligo(dT)-coupled magnetic beads. The improved technique (eRIC, i.e. enhanced RIC) employs a modified probe which allows for more stringent washing steps and therefore reduces background proteins. These include mostly ribosomal RNA. The method is first compared with the existing RIC method using Jurkat cells and in a second experiment quantitative changes are studied in response to DMOG, which is affecting the RNA demethylases. The developed eRIC method could successfully be applied to monitor changes in the RBPome. All experiments are well-designed and sufficiently demonstrate the improvements of eRIC. The manuscript is well-written and clearly describes the results. I do not have major comments. I suggest few very minor changes and corrections before publication of this manuscript at Nature communications.

- The figures are too full. I doubt that everything will be readable at final printing size. As the number of supplemental figures is very low I suggest to move some of the results to supplementary material and/or include more figures in the main text (the maximum number of figures allowed by Nature communications is not yet reached).

We split the two largest figures of the original version (Fig.1 and Fig.4) into two smaller figures each. Consequently, the manuscript now contains 7 main figures instead of 5 before. All figures are now designed in artboards with an A4 page size, and we have confirmed that the content of the figures is readable at this printing size.

- Abstract, line 20: “signal-to noise ratios” change to “signal-to-noise ratios”.

This was corrected accordingly.

- Methods, line 324: “coming from 1 volume” sounds rather colloquial. I suggest to re-phrase.

We replaced the expression for “originated from 1 volume”.

- Methods, lines 342, 346 and 483: “400g” change to “400 x g”.

This was corrected accordingly.

- Methods, line 395: “ON” change to “overnight” (not clear).

This was changed accordingly.

- Methods, line 403: “were employed” change to “was employed”.

This was changed as suggested.